# Neuropilins Controlling Cancer Therapy Responsiveness

**DOI:** 10.3390/ijms20082049

**Published:** 2019-04-25

**Authors:** Virginia Napolitano, Luca Tamagnone

**Affiliations:** 1Cancer Cell Biology Laboratory, Candiolo Cancer Institute-FPO, IRCCS, 10060 Candiolo, Italy; virginia.napolitano@ircc.it; 2Istituto di Istologia ed Embriologia, Università Cattolica del Sacro Cuore, 10168 Rome, Italy; 3Fondazione Policlinico Universitario Agostino Gemelli, 10168 Rome, Italy

**Keywords:** neuropilin, cancer, chemotherapy, radiotherapy, target therapy, immunotherapy, resistance

## Abstract

Neuropilins (NRPs) are cell surface glycoproteins, acting as co-receptors for secreted Semaphorins (SEMAs) and for members of the vascular endothelial growth factor (VEGF) family; they have been initially implicated in axon guidance and angiogenesis regulation, and more recently in cancer progression. In addition, NRPs have been shown to control many other fundamental signaling pathways, especially mediated by tyrosine kinase receptors (RTKs) of growth factors, such as HGF (hepatocyte growth factor), PDGF (platelet derived growth factor) and EGF (epidermal growth factor). This enables NRPs to control a range of pivotal mechanisms in the cancer context, from tumor cell proliferation and metastatic dissemination, to tumor angiogenesis and immune escape. Moreover, cancer treatment failures due to resistance to innovative oncogene-targeted drugs is typically associated with the activity of alternative RTK-dependent pathways; and neuropilins’ capacity to control oncogenic signaling cascades supports the hypothesis that they could elicit such mechanisms in cancer cells, in order to escape cytotoxic stress and therapeutic attacks. Intriguingly, several studies have recently assayed the impact of NRPs inhibition in combination with diverse anti-cancer drugs. In this minireview, we will discuss the state-of-art about the relevance of NRPs as potential predictive biomarkers of drug response, and the rationale to target these proteins in combination with other anticancer therapies.

## 1. Neuropilins: Structure and Functions

Neuropilins are 130–140 kDa single-pass transmembrane proteins. Neuropilin-1 (NRP1) was the first member of the family to be described in 1987 and Neuropilin-2 (NRP2) was isolated later by Chen et al. in 1997 [1,2]. In humans, *NRP1* and *NRP2* genes map to two different chromosomes, 10p12 and 2q34, respectively [3]. Although NRPs share only 44% homology in amino acid sequences, their structure is very similar (Figure 1). They are composed by an extracellular domain, a transmembrane stretch, and a short intracellular tail. The extracellular region contains two complement-like binding domains (a1 and a2), two coagulation factor V/VIII homology-like domains (b1 and b2) and a meprin-like (c) domain [4]. The single transmembrane portion is followed by a short cytoplasmic tail, terminating with a consensus sequence, able to interact with PDZ (PSD-95/Dlg/ZO-1 homology) protein domains. Extracellular “a” and “b” domains are implicated in ligand binding, while the “c” domain mediates neuropilins homo- and heterodimerization, which seems to be essential for function. NRPs were originally identified as coreceptors for class-3 semaphorins (SEMA3s), a family of molecules acting as repulsive or attractive signals for neuronal processes, in a complex with transmembrane receptors type-A plexins [5,6]. Subsequently, NRPs were further characterized as receptors for vascular endothelial growth factors (VEGFs) [4]. Indeed, NRPs are expressed in endothelial cells, where they interact with several members of the VEGF family and some of their tyrosine kinase receptors (VEGF-Rs), enhancing their signaling cascade and promoting angiogenesis. In particular, NRP1 is critical for VEGF-A/VEGF-R2-mediated angiogenesis [7], whereas NRP2 is important for VEGF-C/VEGF-R2/3-mediated lymphangiogenesis [8,9,10]. Although the signaling pathways for NRP1 and NRP2 are generally distinct, they can partially compensate for each other, since the double knock-out mice of both *NRPs* genes displays a more severe phenotype than the single knock-out mice, characterized by the impairment of blood vessel development and early death in utero at E8.5 [11]. Beyond their role in axon guidance and tumor angiogenesis, NRPs have attracted attention for specific functions mediated in cancer cells, largely due to their interaction with other signaling cascades [12]. In particular, NRPs have been found to couple with many other transmembrane receptor molecules, such as epidermal growth factor receptor (EGFR), hepatocyte growth factor receptor (MET), insulin-like growth factor 1 receptor (IGF1-R), platelet-derived growth factor receptors (PDGF-R), tyrosine kinases, transforming growth factor (TGFβ) receptor and integrins, eliciting a range of intracellular signaling cascades [13,14,15,16,17]. Consequently, NRPs have been found to control a range of cellular processes, such as proliferation, survival, invasion and migration. From the mechanistic point of view, it is not fully understood how NRPs can control this range of diverse signaling receptors. It has been shown that NRP1 can regulate the oligomerization on the cell surface of EGFR and the subsequent intracellular signaling [13]. In general, upon the assembly of multimeric signaling complexes, NRPs have been shown to control receptor endocytosis and intracellular trafficking [18,19]. For example, NRP1 can promote the partitioning of VEGF-R2 into vesicles that are recycled back to the cell surface, while in its absence, this receptor tyrosine kinase is targeted for degradation [20]. In human tumors, often upregulation of NRPs expression correlates with poor patient prognosis [21,22,23,24,25]. Here we will focus on the current evidence associating NRPs with cancer responsiveness to conventional and innovative therapies, and their potential implications for precision and targeted oncology.

## 2. Neuropilins and Cancer Responsiveness to Radio- and Chemo-Therapy

Published data suggest that NRPs have a role in cancer response to radiotherapy (RT) and chemotherapy (CT) (see Table 1). Indeed, RT and CT are currently the most common approaches for the treatment of cancer patients, used either alone or in combination [26]. In general, RT and most CT treatments act by targeting genomic DNA, hampering its replication and in turn leading to cell growth inhibition and apoptotic death [27]. Indeed, ionizing radiation (IR) and chemotherapeutic agents induce various types of DNA damage, including strand breaks, DNA adducts and inter- and intra-strand DNA crosslinks; in response, cancer cells activate signaling pathways regulating DNA repair, cell cycle and survival. Among the signal transduction pathways involved in adaptive cancer response to therapy are tyrosine kinase receptors (RTKs) regulated by NRP1, such as EGFR and IGF1-R, and intracellular effectors, such as PI3K/AKT, MAPK/ERK or NF-κB [28,29]. It was therefore speculated the involvement of NRPs in RT and CT sensitivity and resistance mechanisms. In fact, it has been reported that NRP1 silencing in non-small cell lung cancer (NSCLC) Calu-1 cells increases significantly the apoptotic rate and the sensitivity to ionizing radiation (IR), compared to controls. NRP1 appears to regulate RAD51 expression through a VEGFR2-independent ABL-1 pathway, consequently regulating radiation response [30]. It was also found that miR-9 overexpression, targeting NRP1, significantly inhibited the viability and colony-forming ability of NSCLC A549 cells subjected to IR, thereby enhancing their radiosensitivity. Early evidence about NRPs contribution to chemosensitivity was provided in 2005, by Wey et al., who demonstrated that, in pancreatic cancer cells, NRP1 overexpression confers chemoresistance to gemcitabine and 5-fluorouracil, leading to constitutive activation of MAPK signaling, and inhibiting anoikis [31]. Later, a specific antagonist of VEGF binding to the NRP1 b1 domain (EG3287) has been found to increase the cytotoxic effects of chemotherapeutic agents 5-fluorouracil, paclitaxel and cisplatin in lung A549 and prostate DU145 carcinoma cells, through inhibition of integrin-mediated interaction with the extracellular matrix [32]. Moreover, it was shown that NRP1 overexpression in oral squamous cell carcinoma cells CAL27, HN4 and HN6, confers resistance to cisplatin treatment, reducing the apoptotic rate [33]. On the contrary, NRP2 overexpression, mediating SEMA3F signaling, was shown to increase the sensitivity to 5-fluorouracil and oxaliplatin of colorectal adenocarcinoma cells HT29, through the down-regulation of integrin αvβ3 expression [34] (see Figure 2). In a recent paper, the authors explored the involvement of NRP1 in doxorubicin sensitivity of breast cancer cells; they showed that the VEGF-A/NRP1 axis promoted cancer stem cells (CSCs) self-renewal via the Wnt/β-catenin pathway [35]. In the same line of evidence, the VEGF-C/NRP2 axis (through the inhibition of mTOR complex 1) was found to protect prostate and pancreatic cancer cells during chemotherapeutic stress, by activating autophagy to support cancer cell survival [36]. There are two major studies demonstrating that NRPs could actually be considered predictive of therapeutic response. In 2012, in a phase III clinical trial, the authors assessed the levels of plasmatic VEGF-A and the expression of NRP1 and VEGF-R1/2 in locally advanced or metastatic gastric cancer treated with bevacizumab or placebo, in combination with chemotherapy. High plasma VEGF-A and low tumor NRP1 expression turned out to represent strong biomarkers predicting better clinical outcomes in patients with advanced gastric cancer, treated with bevacizumab [37]. By analogy, Keck et al. later showed that assessing NRP2 and VEGF-C expression, by immunohistochemical analysis, allows predicting therapy responsiveness in bladder cancer patients treated with transurethral resection (TURBT) and radio-chemotherapy (RCT) [38]. Altogether, despite functional evidence in vitro about the role of NRPs in the regulation of cancer cell radio- and chemosensitivity, it is fair to conclude that their potential relevance as predictors of clinical outcome in response to RT or CT is not yet established, and the implicated mechanisms deserve further investigation.

## 3. Neuropilins and Cancer Responsiveness to Target Therapies

Targeted therapies are based on the idea of hitting the specific vulnerabilities of certain tumors, for instance cancer cell oncogene addiction [39]. However, due to tumor heterogeneity, certain cells could exhibit an intrinsic resistance to oncogene inhibitors, mediated by the concurrent constitutive activation of additional signaling pathways, which sustain proliferation and survival [40]. Furthermore, upon prolonged drug treatment, even sensitive tumors often recur at a later stage, by developing refractoriness to targeted drugs. The elucidation of the molecular mechanisms behind tumor resistance to therapies will enable us to develop combined treatment strategies aimed at overcoming this major unmet clinical need.

A growing body of evidence shows the involvement of NRPs in the activation of diverse RTKs and other major signaling pathways in cancer, some of which are known to be involved in the resistance to oncogenic inhibitors (Figure 2). For instance, Kim et al. observed that, in pancreatic ductal adenocarcinoma (PDACs), the elevated expression of active integrin β1 leads to the activation of a Src-Akt signaling cascade, and confers resistance to a monoclonal antibody targeting EGFR, cetuximab. Intriguingly, the authors showed that cetuximab resistance could be overcome by co-targeting EGFR and NRP1, using an antibody-based therapy, resulting in the inactivation of an integrin β1-driven bypass pathway [41]. Another interesting study revealed NRP1 upregulation in the adaptive response of prostate cancer (PCa) to androgen-targeted therapy [42]. Using PCa xenograft (LNCaP) model of castration resistant PCa (CRPC), the authors assessed the transcriptional signature representing the adaptive tumor response to androgen dihydrotestosterone (DHT) inhibitor enzalutamide (ENZ), revealing significant upregulation of NRP1 in metastatic samples (mCPRC), compared to localized PCa samples. In vitro validation studies showed that the androgen receptor (AR) signaling may actually suppress NRP1 expression; in fact, NRP1 was found to be upregulated upon treatment with androgen-targeted therapies (ATTs), both in LNCaP xenograft models and in CPRC clinical samples. Finally, the analysis of a PCa patient cohort identified NRP1 as an independent prognostic indicator of early biochemical recurrence (BCR) following radiation therapy. In a phase-II clinical trial in irinotecan-refractory metastatic colorectal cancer (mCRC) patients (BOND-2 study), high intratumoral levels of either EGFR, VEGF-R2 or NRP1 were associated with longer overall survival (OS) of patients receiving targeted combined therapy with monoclonal antibodies cetuximab and bevacizumab (with or without CT), compared with tumors characterized by low EGFR, VEGF-R2 or NRP1 expression [40]. Recently, Rizzolio et al. demonstrated NRP1 function as driver of adaptive resistance to BRAF, HER2 and MET targeted therapies [43]. From a mechanistic point of view, the authors showed that NRP1 induced a signal transduction cascade leading to the activation of a JNK-dependent signaling, which in turn activates SOX2 and JUN transcription factors, that mediate the upregulation of EGFR and IGF1R, respectively, consequently impacting cancer cell growth. Remarkably, the treatment combination with NRP1-interfering molecules improved the efficacy of oncogene-targeted drugs and prevented the onset of resistance (or even reversed it) in multiple tumor models, like melanoma, lung and breast cancer cells [43].

Concerning NRP2, it is of particular interest that a recent work reported that an acquired resistance to MET oncogene-targeted drugs is associated with NRP2 loss in diverse cancer cell models. NRP2 depletion led to NFkB pathway activation and, most notably, upregulated the EGFR-associated protein KIAA1199/CEMIP, known to oppose the degradation of activated EGFR kinase. Indeed, KIAA1199 silencing in oncogene-addicted tumor cells improved therapeutic sensitivity and counteracted the onset of acquired drug resistance [44]. Additional studies are warranted to establish whether NRPs could represent valuable biomarkers to predict the clinical response to target therapies.

## 4. Neuropilins and Therapies Targeting the Tumor Microenvironment

A growing number of anti-cancer therapeutic approaches target non-neoplastic cells in the tumor microenvironment, such as the endothelial and inflammatory cells. In particular, anti-angiogenic drugs and immune checkpoint inhibitors have gained a prominent role in the clinical practice [45,46]. Notably, besides their expression in tumor cells, NRPs are also widely represented in cells of the tumor microenvironment, especially endothelial and immune cells [47]. It is well established that endothelial NRPs are pivotal players in the regulation of tumor angiogenesis, especially as they bind secreted semaphorins as well as members of the VEGF family, and interact in receptor complexes with tyrosine kinase VEGF-R1, R2 and R3 [48,49,50]. The rationale of VEGF inhibition as a tumor suppressor strategy is based on the fact that this factor is a major inducer of angiogenesis, which enables tumor growth, and furthermore provides a route for tumor cells to disseminate distant metastasis. Although VEGF-targeted therapies are now approved for the treatment of metastatic colorectal cancer, advanced lung cancer, renal cell carcinoma, metastatic breast cancer and glioblastoma [51,52,53,54,55], predictive markers for patient selection still remain elusive. Therefore, investigations aimed at identifying biomarkers of responsiveness to anti-VEGF antibodies (bevacizumab) and other VEGF inhibitors need to be carried out. In the AVAGAST trial, it has been shown that the assessment of circulating VEGF-A and tumor levels of NRP1 could aid the selection of patients with advanced or metastatic gastric cancer who are more likely to benefit from the addition of bevacizumab to the chemotherapeutic protocol. Notably, patients with low tumor expression of NRP1 showed a better overall survival rate compared to patients with high NRP1 expression [37]. In other studies, the combination of VEGF inhibitors with NRP1 blocking antibodies (MNRP1685A) appeared to enhance the antiangiogenic effect [56]; however, such protocol led to severe side effects (such as proteinuria and thrombocytopenia) in a Phase 1b study that assayed MNRP1685A combination with bevacizumab [57]. This toxicity was not observed when MNRP1685A was administered alone [57], suggesting that adverse effects might be attributed to a strong synergistic effect on VEGF signaling, also affecting normal vessels.

Immunotherapy with immune checkpoint inhibitors represents a promising improvement in the treatment of diverse cancers. The mechanism of action of these agents is based on the enhancement of effector T cell functions. NRPs expression has been observed in different cells involved in immune response, especially dendritic cells (DC), macrophages and Treg lymphocytes [47]. After exposure to antigens, DCs undergo maturation, migrate to the lymphoid organs and prime naïve T cells to initiate the immune response; actually, NRP1 is involved in the formation of the so-called “immunological synapse” between DC and naïve T cells, an early event necessary for physiological lymphocyte activation [58]. In the tumor microenvironment, however, immune response is subverted in favor of cancer progression. For instance, one of the key players are tumor-associated macrophages (TAMs), recruited from circulating monocytes. Interestingly, NRP1 is well expressed on TAMs and it is critical for their migration to the hypoxic core of the tumor, where they release proangiogenic and immunosuppressive factors [59,60]. Thus, in this context, NRP1 interference was found to inhibit TAM function, promote anti-tumor immune response and inhibit cancer growth and metastatic progression [60]. Actually, cancer cells may evade the control of the immune system by multiple mechanisms, including the recruitment of regulatory T cells (Treg), which serve to establish an immunosuppressive environment. It was found that NRP1 is furthermore required for Treg stability, tumor infiltration and local suppression of immune response [61]. In particular, NRP1-deficient Treg cells were found to release IFN-γ, which in turns suppresses Treg activity, unleashing an anti-tumor immune response [61]. Notably, IFNγ-induced Treg functional inhibition is also a requisite for achieving therapeutic response to anti-programmed cell death protein 1 (PD-1) antibodies, while IFN-γ-Receptor depletion in Treg conferred resistance to anti-PD1 immunotherapy in melanoma mouse models [61]. Thus, NRP1 inhibition could be exploited to hinder Treg-dependent suppression of an anticancer immune response and potentially synergize with immune checkpoint inhibitor drugs. These findings underscore the importance of NRP1 in controlling crucial players of the subverted immune response found in tumors. Therefore, targeting NRP1 could serve to counteract immune suppressive mechanisms for cancer therapy; moreover, the analysis of its expression levels could provide a predictive marker of responsiveness to immunotherapy to be validated in future clinical trials.

## 5. Neuropilin-Inhibitory Molecules for Combined Cancer Therapies

The development of NRP-inhibitory strategies for cancer treatment is an exciting and challenging endeavor. Different approaches have been so far explored: e.g., soluble NRPs (sNRPs), function-blocking monoclonal antibodies (mAbs), peptides/peptidomimetics and targeted microRNAs. Natural soluble forms of NRPs are generated by alternative splicing; they contain the extracellular ligand-binding domains and can function as ligand traps, potentially preventing the interaction of VEGFs and semaphorins with receptors exposed on the cell surface [62]. For instance, sNRP1 inhibited breast cancer cell migration [63] and non-small lung cancer cells invasiveness, in vitro [64]. In Dunning rat prostate carcinoma cell lines, sNRP1 inhibited VEGF165 binding to full-length NRP1, and its overexpression inhibited tumor growth and elicited apoptosis in vivo [62]. A gene-therapy based delivery of sNRP1 in mouse models of leukemia has been found to inhibit angiogenesis and prolong the survival of leukemia-bearing mice [65]. As concerning NRP2, the soluble splice variant s9Nrp2 has been found to interfere with the VEGF-C/NRP2 signaling axis in prostate cancer [66].

Monoclonal antibodies against the VEGF-binding “b” domain of NRP1 seem to potentiate the effects of anti-VEGF therapy. In particular, anti-NRP1A inhibited VEGF-induced migration of human endothelial cells (HUVEC) and tumor formation in mouse models [56]. Moreover, anti-NRP1 antibody was found to inhibit the formation of NRP1/α5β1 integrin complexes, as well as phosphorylation of FAK and p130cas, in breast cancer cell lines [67]. The anti-NRP1 antibody MNRP1685A, which also targets the VEGF-binding domain of NRP1, has furthermore been evaluated in a phase I clinical trial [57]. MNRP1685A was well-tolerated as single agent, but only achieved modest clinical benefit; instead, when co-administered with bevacizumab and paclitaxel, its effectiveness was jeopardized by significant proteinuria as an adverse effect [57]. Monoclonal antibodies targeting the ligand-binding b1/b2 domains of NRP2 have also been developed [68]. They could inhibit the growth of tumor-associated lymphatic vessels in mouse models, likely by interfering with the binding of VEGF-C to NRP2, and decreased the number of metastasis in sentinel lymph nodes and in distant organs [68]. Besides traditional monoclonals, innovative antigen-specific tools called “nanobodies” have been developed to inhibit NRP1 function, and proven to be effective in sensitizing drug-resistant cancer cells to oncogene-targeted therapies [43].

Oligopeptides and peptidomimetics have also been proposed as NRP1-targeting molecules. In particular, developed sequences have been structured to fit the so called neuropilin-binding “C-end Rule” (CENDR) [69], based on the observation that NRPs interact at high affinity with a wide range of molecules containing a basic-charged amino acid sequence at their C-terminus (including class 3 semaphorins, VEGFs and other soluble factors). Some of these small molecules were validated for coupling with the VEGF-binding site of NRPs, preventing ligand-receptor interactions; for instance, EG00229, developed by Jarvis et al., can inhibit VEGF-R2 phosphorylation, leading to increased cancer cell sensitivity to chemotherapeutic agents paclitaxel and 5-fluorouracil [70] as well as to oncogene targeted drugs [43]. However, EG00229 was also found to impair the viability of A549 lung carcinoma cells [70], consistent with their dependence on NRP1 expression, and unrelated to VEGF signaling [13]. The ATWLPPR peptide prevents VEGF165 interaction with NRP1 but not with VEGF-R2, and it has been shown to decrease tumor angiogenesis and tumor growth in vivo [71]. Another compound antagonizing VEGF binding to the NRP1 b1 domain is EG3287, which could reduce VEGF-induced cell migration and sensitize cancer cells to chemotherapy [32]. Notably, a currently ongoing clinical trial is designed to assess the potential efficacy of the 9-amino acid cyclic peptide CEND-1 (also known as iRGD, with sequence: CRGDKGPDC) in combination with nabpaclitaxel and gemcitabine, for the treatment of patients affected by metastatic pancreatic ductal adenocarcinoma (PDAC). This small molecule was found to be rapidly internalized in endothelial cells, and induce vessel permeabilization and CT tissue penetration in vivo [72]. Another potential approach to interfere with NRP function exploits peptides designed to target the transmembrane domain involved in the dimerization of the receptor; this property turned out to be an efficient strategy to inhibit tumor cell growth in mouse models [73].

Finally, NRP1 transcripts are targeted by several microRNAs (miR), such as miR-9, miR-181b, miR-320 and miR-338, with the effect of modulating tumor angiogenesis, invasion and metastasis in experimental models [74,75,76]. In particular, miR-320 was found to impair endothelial cell migration, adhesion and tubule formation, and could inhibit tumor angiogenesis in a model of oral squamous cell carcinoma in mice [75]. miR338 was found to hinder NRP1 expression in melanoma cells, but its levels were downregulated in response to targeted therapy leading to the onset of drug resistance [43]. On the other hand, NRP2 was found to be the target of other miRNAs, namely miR-15b and miR-486-5p. miR-15b could inhibit cell invasion and angiogenic tube formation in a rat glioma cell line (L9) [77,78]. Moreover, it was found that miR-486-5p can act as tumor suppressor in a colorectal carcinoma model, impairing tumor growth and lymphangiogenesis [78]. The prospective application of miRNAs in clinics is challenging, and it requires the use of vehicles, such as microvesicles or liposomes, in order to ensure stability, increase tissue delivery and prevent immune response [79]. Preclinical studies of miRNA therapies using liposomes provided promising results in hepatocarcinoma mouse models [80], but a subsequent clinical trial has been terminated due to immune-related adverse effects [81].

## 6. Conclusions and Open Questions

Neuropilins are versatile transmembrane proteins acting as signaling hubs on the cell surface. Thanks to their ability to take part in diverse growth factor receptor complexes, they can regulate a range of processes relevant in cancer, such as cell viability and proliferation, cell migration, invasion and metastasis formation, angiogenesis and immune response. The multiple assets provided by NRPs to cancer cells could also feature a formidable point of vulnerability, once adequate function-blocking molecules will be validated for application in clinical trials. Published data suggest that an elevated expression of NRPs may identify more aggressive tumors, less responsive to diverse treatment protocols (including CT and RT) and associated with poor disease prognosis. In addition, secondary acquired tumor resistance to therapies (including the most innovative and targeted ones) arises from salvage molecular pathways supporting cancer cell viability, such as those upregulated by NRP1. Indeed, NRP1-targeting proved effective to prevent the onset of drug-resistance or restore responsiveness to oncogene-targeted therapies in melanoma and breast cancer preclinical models. Notably, because of NRPs’ promiscuous interplay with other signaling molecules, the impact of their inhibition in cancer may vary depending on the tumor type, the genetic makeup of cancer cells and the microenvironment context. For instance, NRPs are significantly involved in the regulation of tumor vessels, tumor associated macrophages and lymphocytes. A single clinical trial combining bevacizumab with anti-NRP1 antibodies failed due to enhancement of adverse effects associated with VEGF functional blockade. However, we know that NRPs deploy many more functions relevant in cancer progression beyond VEGF binding, and additional efforts to exploit this knowledge to design combined therapies with NRP-targeted molecules are warranted. In particular, the potential role of NRP1 as a predictive marker for immunotherapy deserves further exploration. In sum, targeting of NRPs appears to be a valuable approach for designing new combined therapeutic strategies. Moreover, a better understanding of NRP-dependent mechanisms controlling cancer progression and responsiveness to therapies deserves further investigation.

## Figures and Tables

**Figure 1 ijms-20-02049-f001:**
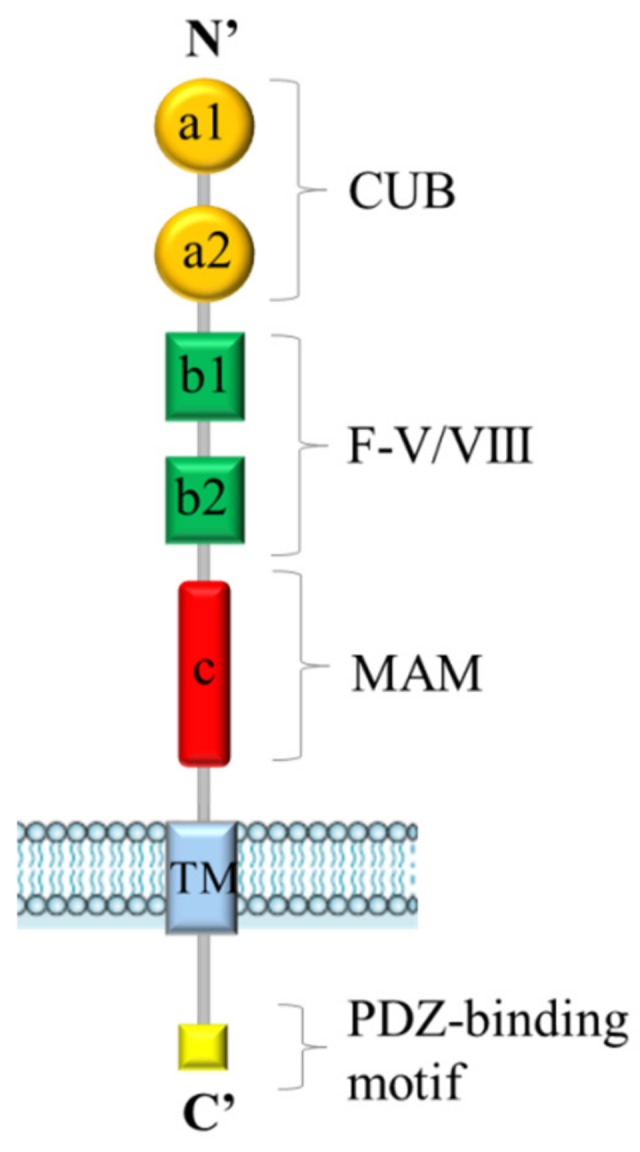
General structure of Neuropilins. Both Neuropilin-1 (NRP1) and Neuropilin-2 (NRP2) contain five extracellular domains (a1/a2, b1/b2 and c domains), a single transmembrane (TM) stretch, and an intracellular PDZ domain-binding motif at C’-terminus.

**Figure 2 ijms-20-02049-f002:**
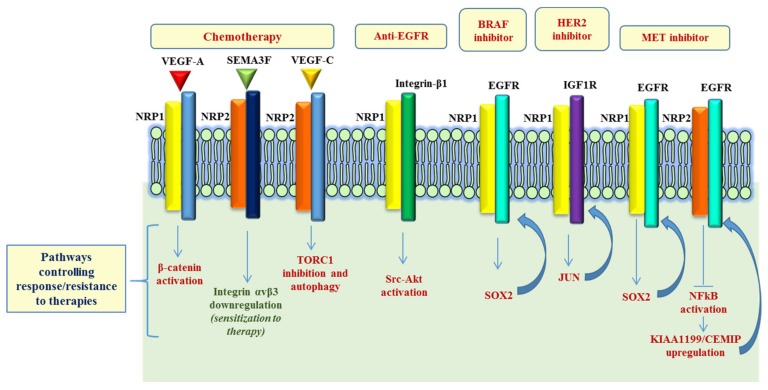
NRP-dependent pathways involved in resistance to therapy. Schematic representation of the signaling pathways regulated by NRPs in response to chemotherapy and controlling responsiveness/resistance to oncogene-targeted therapies. While in most cases, NRPs expression mediates drug resistance, in two studies it was shown that NRP2 negatively controls mechanisms supporting cancer cell viability (e.g., Integrin αvβ3 and KIAA1199) in response to therapy. Refer to main text for details and references.

**Table 1 ijms-20-02049-t001:** Neuropilin levels correlated with responsiveness/resistance to cancer therapies.

Therapy	NRP1 Levels Correlate with	References	NRP2 Levels Correlate with	References
Radiotherapy	low responsiveness (non-small cell lung cancer)	[30]	-	-
Chemotherapy (5-fluorouracil)	low responsiveness (pancreatic, non-small cell lung, and prostate cancers)	[31,32]	high responsiveness (colorectal adenocarcinoma)	[34]
Chemotherapy (platin-derived drugs)	low responsiveness (non-small cell lung, prostate, and oral squamous cell carcinoma)	[32,33]	high responsiveness (colorectal adenocarcinoma)	-
Chemotherapy (paclitaxel)	low responsiveness (non-small cell lung and prostate cancers)	[32]	-	-
Chemotherapy (gemcitabine)	low responsiveness (pancreatic cancer)	[31]	-	-
Chemotherapy (docetaxel)	-		low responsiveness (prostate and pancreatic cancers)	[36]
Chemotherapy (doxorubicin)	low responsiveness (breast cancer)	[35]	-	-
EGFR inhibitor (cetuximab)	secondary resistance (pancreatic cancer)	[41]	-	-
Androgen-targeted therapies (ATTs)	secondary resistance (prostate cancer)	[42]	-	-
cMET inhibitor (JNJ38877605)	secondary resistance (gastric and lung cancer)	[43]	high responsiveness (lost upon acquired resistance) (gastric and lung cancer)	[44]
B-Raf inhibitor (PLX-4720)	secondary resistance (melanoma)	[43]	-	-
Her2 inhibitor (lapatinib)	secondary resistance (breast cancer)	[43]	-	-
anti-VEGF (bevacizumab)	low responsiveness (gastric cancer)	[37]	low responsiveness (bladder cancer)	[38]

Table 1 lists reported correlations between high expression levels of NRP1 or NRP2 and responsiveness or resistance to different kinds of cancer therapies. Tumor samples derived from treated patients were analyzed in part of the studies (by immunohistochemistry or mRNA qPCR); in other cases, gene expression was engineered in cultured cancer cells subjected to therapy. In parentheses, the implicated drugs (in first column) and tumor types (in second and fourth column) are specified; “-“ stands for “not reported”.

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
