# Peer review of "Neuropilins Controlling Cancer Therapy Responsiveness"

_ijms, 2019, doi:10.3390/ijms20082049_

Round 1
Reviewer 1 Report
The authors review in their manuscript the current state of knowledge how neuropilins, being important regulators of oncogenic signaling cascades, are associated with responsiveness to conventional and innovative cancer therapies, and they point to novel ways to improve conventional therapies by incorporating neuropilin targeting strategies.
The relevant literature has been comprehensively researched and analyzed / evaluated. The manuscript is well written, and all statements and conclusions are well documented and convincingly substantiated.
Yet, there are a few comments to be made:
Specific comments and suggestions:
Lines 76 ff.: Why is Wnt signaling not mentioned here along with the pathways regulating the adaptive response to therapy, while in line 100, the influence of NRP on Wnt signaling is mentioned?
For Table 1, a descriptive title and an explanatory legend would be helpful to quickly grasp what is gathered here. The ‘-‘ symbols in the table should be explained. Has a correlation not yet been investigated, or did the therapy outcome not correlate with the neuropilin levels? Have the neuropilin levels been determined at the protein or at the mRNA level?
Lines 105 ff., 142 ff., and 236: Why are the clinical trials mentioned here (references 38, 39, 44, 75) not included in Table 1?
At ClinicalTrials.gov, currently eight studies are found when searching for “neuropilin” AND “cancer”. In particular, would not the phase I trial on the effect of neuropilin-targeting CEND-1 in combination with Nab-paclitaxel and gemcitabine be of interest for this review?
Line 184: Proteinuria has been observed as a severe side effect, when MNRP1685A had been administered together with bevacizumab. Were also adverse drug reactions observed with other medications or treatments that affect neuropilin levels, and if so, what are the exact side effects?
Line 267: While packaging of miRNAs in microvesicles (not ‘microvescicles’) for therapeutic use is possible, encapsulation into exosomes appears rather complicated. In reference 85 (author names should read ‘Hydbring P and Badalian-Very G’, respectively), exosomes are discussed rather as a diagnostic source of miRNAs and as a means to deliver viral RNAs, respectively (Pubmed IDs 19627513 and 20304794).
Line 269. At this point, the authors should emphasize that caution should be necessary in this context: The phase I study of miR-RX34 liposomal injection (NCT0129971, reference 86) has been terminated due to five immune-related serious adverse events (cf.: https://www.clinicaltrials.gov/ct2/show/NCT01829971?term=nct01829971&rank=1).
Minor points:
A list of abbreviations would be helpful and desirable. For example, what does AR (line 138) stand for? Also, abbreviations should be consistent, e.g., either VEGF-R2 (line 46) or VEGFR2 (line 84) and MNRP1685A (line183) or MNRP1685-A (line235).
Each statement should ideally be accompanied by a reference. Sometimes the corresponding reference is mentioned much later, e.g., the work of Rizzolio is first mentioned in line 147, while it is cited not before line 154. Or, in line 219 among others, NRP-targeting miRNAs are mentioned, but respective references are not given before line 258.
Line 31: By convention, human gene symbols should be all uppercase, italicized, and without Roman, Greek or punctuation symbols, i.e., NRP1 and NRP2 for neuropilin-1 and -2, respectively. Human protein symbols should be identical but not italicized, i.e., NRP1 and NRP2. For mice, however, gene symbols should be italicized with only the first letter in uppercase (Nrp1) and protein symbols not italicized but all letters in uppercase (NRP1).
Line 164, 168: the various non-neoplastic cell types as well as the ‘certain’ immune cells could be specified at these sites, respectively.
Line 194/195: Is there a reason why the terms "immunologic synapse" and 'immune synapse' are used in parallel?
Few minor grammar errors, e.g., lines 141, 240/1, 249
The references should be carefully proofread, and formatting of the references should be checked. ‘s.l.’ appears unnecessary.
Some references are mere duplicates: references 10 (line 47) = 77 (line 241), 57 (Table1, line 183) = 73 (line 232), 58 (line 185) = 76 (line 238), 59 (line 186) = 75 (line 236), and 67 (line 222) = 68 (line 225). Accordingly, the statements in line 182 ff. appear essentially to be repeated in lines 234 ff.
Reference 68: The title is missing. However, it seems to be a duplication of reference 67.
Two consecutive periods at the end of the reference (e.g., reference 19)
Inconsistent capitalization of titles, e.g., references 28, 35-38
Journal abbreviation should be used throughout (e.g. reference 31, 32)
Reference 43: ‘al., Tse BWC et.’ should be ‘Tse BWC et al.’, but rather than ‘et al.’, authors’ names should better be specified.
Journal name abbreviation without periods (references 58, 59)
Journal name not in all-caps (reference 38, 44)
Page numbering should be consistently abbreviated or full length. (e.g., references 67 and 68)
Correct spelling of the titles (e.g., references 71, 75, 80)
Reference 76: ‘Jarvis A’ instead of ‘JJarvis A’
Reference 86: Bibliographical information of the trial (to which the link refers) is missing
Reference 87: The doubled ‘et al., et al.’ should be replaced by the names of the authors.
Specifying all Pubmed IDs would be helpful.
Reviewer 2 Report
I reviewed the review article by Napolitano and Tamangone titled "Neuropilins Controlling Cancer Therapy Responsiveness". The manuscript is well written and organized and provides a comprehensive summary of the reported functions of neuropilins in cancers and their role in resistance to chemo- and radiation therapy. The readers would benefit more if schematic illustrations of structure and functions of neuropilins and the reported mechanism of action, to highlight the potential benefits of targeting neuropilins.
Author Response
Napolitano et al.
Responses to Reviewer’s Comments
REVIEWER 2
I reviewed the review article by Napolitano and Tamangone titled "Neuropilins Controlling Cancer Therapy Responsiveness". The manuscript is well written and organized and provides a comprehensive summary of the reported functions of neuropilins in cancers and their role in resistance to chemo- and radiation therapy. The readers would benefit more if schematic illustrations of structure and functions of neuropilins and the reported mechanism of action, to highlight the potential benefits of targeting neuropilins.
Response: We thank the Reviewer for these insightful suggestions. We have thereby included two new Figures 1 and 2 in the revised manuscript.
Reviewer 3 Report
The current manuscript is a mini-review on the relevance of neuropilins as potential biomarkers in cancers and on the rationale to target these molecules in anti-cancer therapies. Overall, the manuscript is well prepared with care and attention to detail.
The main concern was related to the description of the roles of Nrp in cancer immune tolerance, which could be confusing to the readers (p. 5).
- Line 189: the authors wrote that inhibitors of immune checkpoints enhance effectors T cell functions, which is promising to improve treatment of diverse cancers. It is now well-known that blocking the functions of immune checkpoint proteins, such as PD-L1 on tumor cells and PD-1 on T cells, with a so-called immune checkpoint inhibitor (anti-PD-L1 or anti-PD-1) allows effectors T cells to kill tumor cells. However, the authors said that Nrp-1 is involved in the activation of naïve T cells, and that blocking Nrp-1 decreases T cell proliferation. Accordingly, anti-Nrp1 molecules would help cancer cells to evade the control of immune system, through the inhibition of effector T cell activation. The authors have to clarify this point. Moreover, it is unclear whether Nrps are directly involved in the mechanisms of action of immune checkpoints.
- Line 208: Nrp-1 is required for Treg stability. Because these cells are involved in the establishment of immunosuppressive environment, targeting Nrp-1 could be a way to counteract immune suppressive mechanisms. However, the authors wrote in the same paragraph that IFN-induced Treg fragility is a prerequisite therapeutic response to anti-PD1 antibodies. The relations between Treg and PD1 is unclear. Moreover, they said that “these effects (= Treg stability, tumour infiltration and local suppression of immune responses?) were found to be mediated by IFN release from Nrp1-/- Treg cells “. What does it mean? IFN is involved in Treg fragility, but not in Treg stability and functions?
The authors have to rephrase this part of the review.
Minor points:
- the authors should provide a figure representing the structure of both Nrp1 and Nrp2 in the first part of the review “structure and functions”. This figure would be helpful for the readers to visualize the domains within Nrp structures (see part; 5, p. 6).
- according to the instructions for authors, reference numbers in the text should be placed in square brackets [ ], as follow [1], [1–3] or [1,3]; and references should be described as follows in the reference list: 1. Author 1, A.B.; Author 2, C.D. Title of the article. Abbreviated Journal Name Year, Volume, page range. doi.
Reviewer 4 Report
The authors present a nice and readable mini-review on an interesting topic. The manuscript successfully gives an overview of the current state-of-affairs regarding neuropilins and their involvement in cancer and cancer treatment. I have only some minor comments to further improve the manuscript.
Major comments:
An schematic illustration that depicts the structure of NRP-1 and NRP-2 as well as the pathways they may regulate in a cancer setting will support the text and make the manuscript more appealing.
Minor comments:
Abstract, line 9 – Please remove the phrase “non-tyrosine kinase” as this may confuse readers. Neuropilins are cell surface glycoproteins not kinases.
Page 2, line 77 – One cannot state that a total of four pathways participate in the regulation of adaptive cancer response to therapy. There may be additional pathways involved. To solve this issue one could say: “…..TGF-β, frequently participate…….”
Page 2, line 79-81 – Please rephrase, this sentence (“Considering……..) as it is not clear what the author try to convey.
Table 1 – Is the Table 1 comprehensive? Why are refs 38 and 39 not included in the Table?
Section 4, page 4-5 – Please introduce a subheading in this chapter to make it easier to follow.
Page 6, line 236-238 – Please rephrase, sentence not clear.
Page 6, line 238 -241 - Please rephrase, sentence not clear.
Page 6, line 247-250 - Please rephrase, sentence not clear.
Reference 68 – Title is missing, please correct.
